# Chemical Characterization and Quality Assessment of Copaiba Oil-Resin Using GC/MS and SFC/MS

**DOI:** 10.3390/plants12081619

**Published:** 2023-04-11

**Authors:** Joseph Lee, Mei Wang, Jianping Zhao, Zulfiqar Ali, Mohammed F. Hawwal, Ikhlas A. Khan

**Affiliations:** 1National Center for Natural Products Research, School of Pharmacy, University of Mississippi, University, MS 38677, USA; 2Natural Products Utilization Research Unit, Agricultural Research Service, U.S. Department of Agriculture, University, MS 38677, USA; 3Department of Pharmacognosy, College of Pharmacy, King Saud University, Riyadh 4545, Saudi Arabia; 4Division of Pharmacognosy, Department of BioMolecular Sciences, School of Pharmacy, University of Mississippi, University, MS 38677, USA

**Keywords:** adulteration detection, chemical characterization, chromatography/mass spectrometry, copaiba oil-resin, quality assessment

## Abstract

In recent years, the popularity of copaiba oil-resin has increased worldwide due to its medicinal value and wide applications in industry. Despite its popularity, the oil has not been standardized by industry or regulatory agencies. Product adulteration in order to maximize profits has become a problem. To address these issues, the current study describes the chemical and chemometric characterization of forty copaiba oil-resin samples by GC/MS. The results demonstrated, with the exception of commercial samples, that all sample groups contained six characteristic compounds (β-caryophyllene, α-copaene, trans-α-bergamotene, α-humulene, γ-muurolene, and β-bisabolene) in varying concentrations. Furthermore, compositional patterns were observed in individual groups which corresponded to sample origin. Within the commercial group, two samples did not contain or contained only one of the characteristic compounds. Principal component analysis (PCA) revealed distinct groups which largely corresponded to sample origin. Moreover, commercial samples were detected by PCA as outliers, and formed a group far removed from the other samples. These samples were further subjected to analysis using a SFC/MS method. Product adulteration with soybean oil was clearly detected, with each individual triglyceride in soybean oil being unambiguously identified. By combining these analytical techniques, the overall quality of copaiba oil-resin can be assessed.

## 1. Introduction

Obtained from members of the genus *Copaifera*, copaiba oil-resin has a long history of use as a traditional medicine and is one of the most widely used phytomedicines in Brazil. *Copaifera* trees are slow-growing plants which can reach heights of up to 40 meters and grow up to 4 meters in diameter. The trunks of the trees, which contain oleoresins, are dark and have a rough texture. Interestingly, these trees can live to be 400 years old [1,2,3]. More than seventy species of *Copaifera*, which is also collectively known as “Copaiba”, grow primarily in Central and South America; however, four species can be found in Africa. Perhaps the most biologically diverse grouping of *Copaifera* can be found in the northern region of Brazil, which is home to twenty-six species and eight varieties [2]. 

Currently, copaiba oil-resin remains a popular and widely used phytomedicine, particularly in South America [1,2,3]. Traditionally, copaiba oil has been used for the treatment of skin disorders and inflammation [3,4,5]. This use was purported to have stemmed from observations that injured animals rubbed their wounds on the trunk of the *Copaifera* tree [5]. Although primarily used for its purported anti-inflammatory, antiseptic, and wound-healing properties, the oleoresin has also been used to treat a variety of urinary, skin, rheumatic, and respiratory conditions by practitioners of traditional medicine [1,2,3,4,6,7,8]. In addition to its use as a traditional medicine, the oleoresin has also been used in a wide range of cosmetic and pharmaceutical preparations such as soaps, perfumes, ointments, and oral products [2]. 

Although many members of *Copaifera* are used for oleoresin production, up to 70% of commercial oleoresin is obtained from *C. reticulata*. In addition to *C. reticulata*, *C. multijuga* and *C. langsdorffii* are the most common species found in South America [1]. The yellow to light brown oleoresin is obtained through a perforation in the trunk of the tree and is often mixed with additional oleoresin obtained from several trees. This mixture, which is composed primarily of non-volatile diterpenes and volatile sesquiterpenes, may include oleoresin collected from several species of *Copaifera* [1,2]. Sesquiterpenes, which can comprise more than 90% of the oleoresin, generally contain a mixture of the following: α-humulene, β-caryophyllene, caryophyllene oxide, α-cadinol, Δ-cadinene, β-elemene, β-bisabolene, α-cubebene, trans-α-bergamotene, α-selinene, and β-selinene [1,2,6,9,10,11,12]. In addition to sesquiterpenes, minor amounts of diterpenes with either kaurane, clerodane, or labdane-type skeletons can also be found in the oleoresin [6]. Since the chemical composition of oleoresin from different species can vary, the standardization of *Copaifera* resin is challenging. For example, oleoresin collected from *C. officinalis* has been reported to contain up to 87% β-caryophyllene, in contrast to oil-resin collected from *C. langsdorffii* which has been reported to contain only up to 33% β-caryophyllene [1]. Since different species of *Copaifera* often grow within the same area, oil-resin is collected from multiple species and combined for commercial use. Although the chemical composition of the oil-resin mixture varies from that obtained from a single species, this is not considered to be adulteration [5,7,9,13,14,15,16].

Despite its increasing popularity, copaiba oil-resin remains unstandardized by regulatory agencies or by industry [2]. In addition, the ISO has no published guidelines for the production and testing of copaiba oil. Due to the time-consuming and labor-intensive collection process, the supply of oil-resin is often limited. Product adulteration often occurs in order to reduce product cost and to maximize profits [7]. Two methods of oil-resin adulteration have been reported: (1) adding either mineral or vegetable oil to authentic resin, or (2) adding cheaper essential oils of other plants which are similar in odor and chemistry [5,17]. For example, the substitution of wood oil (also known as *gurjun balsam* in India) has been reported [18]. In order to address these concerns, a variety of analytical methods have been proposed.

A simple method to detect copaiba oil adulteration was developed using the oil’s refractive index and its thin layer chromatography (TLC) profile [5]. This method was particularly useful in the detection of adulteration by the addition of vegetable oil. Although vegetable oil is a common adulterant found in commercial samples, this method could not identify the exact vegetable oil and did not assess any other aspects of copaiba oil, such as chemical composition. For standardization and pharmacological purposes, the chemical composition remains important. Sousa and colleagues developed and validated a GC/FID method to quantify three sesquiterpenes (β-caryophyllene, α-copaene, and α-humulene) in copaiba oil for standardization purposes [19]. This method was not only robust, but also demonstrated reproducible results. Unfortunately, this method only used three compounds to assess the quality of the copaiba oil. In addition, the samples were collected from only one species (*C. langsdorffii*) from one geographic area. Although chemical composition among *Copaifera* species does not vary widely, the concentration of specific compounds can have considerable variation. In addition, compound concentration variations can be found within the same species growing in different locations. It is unclear if this method could be applied to many commercial samples, since oil-resin obtained from several trees is often mixed together during production [5,9,11,13,14,15]. 

Chemometric analysis can be a powerful tool for the analysis of essential oils. This technique has been used to provide insights into essential oils’ chemical composition, purity, and potential therapeutic applications [20,21]. The use of chemometrics in the analysis of copaiba oil can help to identify the various chemical components of the oil, quantify their concentrations, and determine the quality and purity of the oil. Additionally, chemometric techniques can help identify potential adulteration or contamination of the oil, which is important for ensuring the safety and efficacy of the oil for its intended applications.

Currently, the lack of an established standard for the chemical composition of copaiba oil-resin being sold to consumers poses both a health and safety risk. With this in mind, our goal was to evaluate the chemical composition and variation of samples obtained from different geographic regions. With this data, we hope to establish a chemical profile and identify marker compounds which can be used to establish quality standards for copaiba oleoresin. An additional goal includes identifying possible sources of adulteration. Overall, with the establishment of quality control methods and standards in combination with chemometric analysis, both the health and safety of consumers can be enhanced. 

## 2. Results and Discussion

### 2.1. Characterization of the Chemical Composition of Copaiba Oil-Resin by GC/MS Analysis 

A total of 40 copaiba oil-resin samples were collected for the present study. Among them, 21 samples were from Brazil of which five samples were authentic samples, while the authenticity of 16 samples were unknown. Out of the 16 samples, a specific location of origin was known for 12 of the samples. The remaining four samples did not have a specific origin listed. In addition to samples from Brazil, four commercial samples were obtained from Peru and two from Ecuador. For thirteen samples, no information regarding the country of origin or authenticity was provided. 

The GC/MS total ion chromatogram (Figure 1A) illustrates the occurrence of characteristic compounds common to all of the authentic samples. For example, β-caryophyllene is the primary compound ranging from 38.58–45.77% in this sample group, with α-copaene being the second most abundant compound ranging from 7.33–11.25%. Other major compounds found in this group include trans-α-bergamotene, α-humulene, γ-muurolene, and β-bisabolene. For the most part, the unknown samples (Figure 1A) follow the compositional pattern set by the authentic samples. However, composition variation is greater in this group as illustrated by the wider range of β-caryophyllene concentration found in these samples (35.49–67.71%). Samples in which the location of origin was known could clearly be distinguished by their total ion chromatograms (Figure 1B). In addition, compositional variations were also evident between origin locations. For example, samples from Labrea possessed the greatest amount of β-caryophyllene (53.74–56.87%) when compared to the other location groups. In contrast, samples from Apui contained only 11.61–12.57% β-caryophyllene; however, this group did contain the greatest amount of trans-α-bergamotene (13.65–14.24%). Samples from Tapaua and Parintins shared many compositional range similarities of major compounds such as β-caryophyllene. Interestingly these six compounds, viz., β-caryophyllene, α-copaene, trans-α-bergamotene, α-humulene, γ-muurolene, and β-bisabolene (structures shown in Figure 2), were present in all sample groups except the commercial sample group. Within this group, the six compounds were undetected in two samples; while other commercial samples possessed these six compounds, they did not follow the general compositional patterns of other groups. For example, the composition of α-copaene varied widely in commercial samples (0–32.93%), while the range was smaller for all other sample groups (3.55–11.96%). It was also noted that the compound γ-muurolene was present in all sample groups (ranging from 1.06–4.06%) with the exception of the commercial samples. The commercial samples all had less than 1% of γ-muurolene. In addition, several late eluting compounds (*R_t_* > 60 min), such as manool, kolavenol, and methyl kolavenate were detected in some of the samples with relatively low concentrations. Among these compounds, methyl kolavenate was absent or trace in authentic and unknown samples (<0.52%), but was high in some commercial samples: #619 (2.47%), #628 (3.68%), and #877 (1.67%), respectively. Although these compounds may be derived from copaiba oil-resin, the low concentrations of the compounds limit their use as biomarkers for copaiba oil-resin. Tentative compound identification and the range of composition (average) in each group of samples are given in Table 1. The major compounds detected in each sample along with their compositions are given in Appendix A. The comparison of major compounds in each group is illustrated in Figure 3. 

### 2.2. Chemometric Analyses

After processing the samples using the AMDIS program, the raw data was exported to SIMCA-P+13.0 software. The SIMCA-P software provides multiple tools and options which allow the user to create a custom analysis of a given data set. Perhaps one of the most useful tools is principal component analysis (PCA). It can be used to group samples according to chemical similarity. Figure 4 illustrates the PCA generated by the copaiba oil sample analysis.

Upon further examination, distinct clusters of samples are evident. For example, all samples from Apui form a group located in the upper portion of the PCA plot, indicating a strong similarity between samples within this group. However, this group’s distance from other groups also indicates variations in Apui samples’ chemical composition in comparison to other sample groupings. Likewise, samples from Tapaua, Parintins, and Labrea each form distinct groupings as well. Upon further examination, it is evident that both the Tapaua and Parintins groups are clustered together, indicating that the samples are very similar. This clustering agrees with the compositional patterns observed in the GC/MS sample analysis. Both the authentic and unknown samples, with the exception of three unknown samples (#327, #621, #863), form a group indicating the similarity of their compositions. The commercial samples form a distinct grouping which is removed from the other samples in the PCA. This is consistent with the GC/MS analysis, which indicated irregularities within this group. 

### 2.3. Detection of Adulteration and Identification of Adulterants

Due to the time-consuming and labor-intensive process of collecting copaiba oil, adulteration in order to maximize profits is a common occurrence. According to the literature, two of the most prevalent methods of adulteration include: (1) adding cheaper essential oils to copaiba oil or (2) adding vegetable oil in order to dilute copaiba oil [5,17]. While the addition of cheaper essential oils can often be detected using GC/MS, the detection and unambiguous identification of vegetable oil requires an alternative technique [22,23,24]. One method used to identify vegetable oil involves the identification of triglycerides, which are major compounds present in vegetable oils [25]. Both normal and reversed phase liquid chromatography (NP-HPLC and RP-HPLC) analytical methods have been reported; however, these methods have disadvantages, such as the use of costly solvents and long run-times, usually 1–2 h [23,24,25,26,27]. Although the analysis and determination of triglycerides using GC has been reported, some methods require high temperatures which can lead to compound and column degradation [23,28,29]. Another method involves the transesterification of triglycerides to their corresponding fatty acid methyl esters (FAMES), which can be easily analyzed by GC/MS. Unfortunately, this method cannot be used for the unambiguous, direct identification of triglycerides [23,24]. An alternative analytical technique is supercritical fluid chromatography coupled with mass spectrometry (SFC/MS). Not only can this method enable the unambiguous identification of triglycerides, but it also provides several advantages over other analytical techniques. For instance, the typical analysis of a sample can be accomplished in under fifteen minutes. Since the mobile phase is primarily composed of carbon dioxide, the technique does not require the use and disposal of large amounts of costly solvents when compared to conventional LC methods. Compound degradation due to high temperatures, as in GC, is also avoided with this technique [24,28].

#### 2.3.1. Unambiguous Identification of Triglycerides Using SFC/MS

SFC/MS is a powerful technique which can be used for the unambiguous identification of triglycerides (TGs) [24,28,30]. Fragmentation patterns produced by ESI and in-source collision-induced disassociation (IS-CID) provide structural information which is necessary for confident identification [24]. For example, the most abundant ion, [M+NH_4_]^+^, provides structural information which is important for TG identification. In addition, IS-CID can produce additional ions ([M+H-RCOOH]^+^) which can be used to provide additional information to aid in the accurate identification of TGs based upon the formation of acyl ions. Chemically, TGs are complex hydrophobic molecular species formed by the esterification of three fatty acids (FAs) with a glycerol backbone under enzymatic catalysis. Figure 4 illustrates the formation of the fatty acid from the sn-1, sn-2, and sn-3 positions on the glycerol backbone by SFC/MS with ESI positive ionization. The loss of a fatty acid at the sn-2 position results in the formation of a five-member ring. In contrast, the loss of a fatty acid at the sn-1 or sn-3 position results in the formation of a six-member ring. Since the six membered ring is relatively more stable than the five membered ring, cleavage from the sn-1 and sn-3 position is preferred when compared to sn-2 [24,30,31,32]. A possible mechanism illustrating the favorable loss of a fatty acid from the sn-1 or sn-3 position is presented in Figure 5.

For instance, a TG composed of three identical acyl groups (R_1_R_1_R_1_) attached to the glycerol structure can only produce one ion [R_1_R_1_]^+^, as illustrated in Figure 6 by OOO (triolein). TGs composed of two different acyl groups (R_1_R_2_R_1_ or R_1_R_1_R_2_) can yield two different ions ([R_1_R_1_]^+^ and [R_1_R_2_]^+^). For example, LLO (1,2-linolein-3-olein) is fragmented to form both [LL]^+^ and [LO]^+^ ions, while TGs composed of three different acyl groups (R_1_R_2_R_3_) form three unique ions, namely [R_1_R_2_]^+^, [R_1_R_3_]^+^, and [R_2_R_3_]^+^. In Figure 6, the TG PLO (1-palmitin-2-linolein-3-olein) forms ions [LP]^+^, [OP]^+^, and [LO]^+^, which should theoretically have a 1:1:1 abundance ratio; however, this is not the case. A possible mechanism which can explain the difference between the measured and theoretical abundance ratio is illustrated in Figure 5. As a result of the preferential loss of acyl groups at the sn-1 and sn-3 position, the identification of the sn-2 acyl group can be identified by examining fragment ions in the mass spectra of the TG. As an example, the mass spectra of PLO (Figure 6) include [LP]^+^, [OP]^+^, and [LO]^+^ fragment ions; however, the abundance of [OP]^+^ is the smallest compared to that of [LP]^+^ and [LO]^+^. From the smallest ion abundance of [OP]^+^, one can deduce that the corresponding acyl group was located in the sn-2 position of the TG [23,24,30,31]. 

#### 2.3.2. Detection of Soybean Oil in Commercial Copaiba Oil-Resin Samples

With the unambiguous identification of TGs established, an unknown vegetable oil adulterant can be identified. In the chromatogram of Figure 7, the individual TGs present in soybean oil were identified using the previously described SFC/MS method. When suspect copaiba oil samples were analyzed using this method, a similar chromatographic pattern emerged. The MS data further indicated that TGs were present which were identical to those in soybean oil. This demonstrated that all of the commercial samples had been adulterated with soybean oil, as reported in the literature [5,17]. Not only can this method be used to detect adulteration, but it can also be used to identify the type of vegetable oil which was used as an adulterant.

## 3. Materials and Methods

### 3.1. Copaiba Oil-Resin Samples

In total, forty copaiba oil-resin samples were subjected to analysis. Of the forty samples, twenty-one were sourced from Brazil. Within the Brazilian group, five samples (#613, #614, #615, #616, #860) were authenticated (passed third-party quality testing at the time of collection), while the authenticity of sixteen samples were unknown. For twelve of the samples, a specific location of origin could be assigned. The locations included: Tapaua (#888, #889, #890), Apui (#891, #892, #893), Parintins (#894, #895, #896), and Labrea (#897, #898, #899). The remaining four samples (#617, #618, #624, #629) did not have the specific location of origin listed. Aside from Brazil, four commercial samples were obtained from Peru (#619, #628, #862, #877) and two from Ecuador (#620, #874). For thirteen samples (#327, #621, #622, #623, #625, #626, #630, #863, #865, #871, #873, #878, #880), no information regarding the country of origin or authenticity was provided. The detailed sample information is given in Table 2.

### 3.2. Chemicals and Reagents

HPLC grade dichloromethane, methanol, and isopropanol were purchased from Fisher Scientific (Pittsburgh, PA, USA). The internal standard (IS), C_13_H_28_ (n-tridecane), was obtained from the Polyscience Corporation (Niles, IL, USA). The reference standards of α-humulene, β-caryophyllene, caryophyllene oxide, β-elemene, and soybean oil were purchased from Sigma-Aldrich (St. Louis, MO, USA). LC-MS grade ammonium acetate, ammonium formate, and formic acid were also purchased from Sigma-Aldrich. Beverage grade CO_2_ was purchased from nexAir (Memphis, TN, USA).

### 3.3. Sample Preparation

Before analysis, each copaiba sample was diluted in dichloromethane (0.01%, *v*/*v*), and n-tridecane (IS) was added to each diluted sample solution at a constant concentration of 300 μg/mL.

### 3.4. Gas Chromatography/Mass Spectrometry (GC/MS) Analysis

Samples were analyzed using an Agilent 7890 GC (Agilent Technologies, Santa Clara, CA, USA) equipped with a 7693 autosampler. The separation was achieved on an Agilent DB-5MS ultra inert column (60 m × 0.25 mm × 0.25 μm). The helium carrier gas was set to constant flow mode at 1 mL/min. The inlet was held at 260 °C and was operated in split mode with a split ratio of 50:1. The GC oven temperature was ramped from 80 °C at 3 °C/min to 125 °C, programmed at 1 °C/min to 140 °C, held for 10 min at 140 °C, then ramped at 3 °C/min to 170 °C, and finally ramped at 8 °C/min to 280 °C, where it was held for 10 min. Triplicate injections of each sample were made with a volume of 1 μL.

The mass spectral detector was an Agilent 5977A quadrupole mass spectrometer operated in the full spectral acquisition mode. The mass spectrometer was equipped with an electron ionization source, which was operated with an electron voltage of 70 eV. The ion source, quadrupole, and transfer line temperatures were set to 230, 150, and 280 °C, respectively. Data was acquired using MassHunter Acquisition software (B.07006.2704). Data analysis was performed using MassHunter Qualitative analysis software (B.07.00). Compound identification involved a comparison of the spectra with the NIST database (Version 2.2) using a probability-based matching algorithm. Further identification was based on the relative retention indices compared with the literature [33]. 

### 3.5. Supercritical Fluid Chromatography/Mass Spectrometry (SFC/MS) Analysis

All of the analyses were carried out using a Waters ACQUITY UPC^2^ system (Waters Corporation, Milford, MA, USA). Full spectra for electrospray ionization (ESI) in the positive mode were recorded in the range of 100–1200 amu. The chromatographic separation was achieved using a Waters ACQUITY UPC^2^ Torus 2-PIC column (130 Å, 1.7 μm particle size, 3.0 mm × 150 mm). Data acquisition and processing were performed using MassLynx (Version 4.1 SCN 805) software. The mobile phase consisted of CO_2_ as solvent A and 10 mM ammonium acetate in isopropanol as solvent B. The initial conditions were 1.0% B, linearly programmed to 6.5% B over 10 min, then increased to 30.0% in the next 1 min. After maintaining this condition for 2 min, the column was set to the initial condition for 1 min and then re-equilibrated for 6.5 min prior to the next injection. The flow rate was 1.0 mL/min. The injection volume was 1 μL. The column and autosampler temperatures were maintained at 50 °C and 10 °C, respectively. 

Mass spectrometry was performed using a Waters ACQUITY single quadrupole mass spectrometer. The ESI source was operated in positive ionization mode for scanning. The capillary and cone voltages were 3 kV and 60 V, respectively. The temperature of the source was 120 °C and desolvation gas temperature was set to 450 °C. The cone and desolvation gas flows were 25 L/h and 500 L/h, respectively. A post photodiode array splitter was installed before the MS electrospray probe, which allowed the introduction of a liquid make-up flow to assist the ionization. The make-up flow, composed of methanol with 8 mM ammonium formate and 0.5% formic acid, was delivered by a Waters 515 binary pump at a flow rate of 0.4 mL/min. The ABPR pressure was 2000 psi. 

### 3.6. Statistical Analyses

Extraction of the GC/MS data was performed using the NIST Automated Mass Spectral Deconvolution and Identification Software (AMDIS), Version 2.2. A holistic, non-targeted approach was used for the preprocessing of GC/MS data using SIMCA-P+13.0 software (Umetrics AB, Umeå, Sweden) with the variables in the dataset being Pareto scaled; then, principal component analysis (PCA) was performed. The internal standard selected for GC/MS analysis was used for the alignment and normalization of the peak intensities.

## 4. Conclusions

Although a variety of analytical techniques have been used to assess the quality of copaiba oil-resin and to detect adulteration, such as TLC and GC/FID/MS, a comprehensive investigation of a large sample set has been lacking [5,8,10,17,19]. In an effort to aid in the development of copaiba oil-resin quality standards and to detect adulteration, a combination of analytical techniques was employed. For example, the analysis of a large sample set by GC/MS produced a general chemical profile, while also providing compositional ranges of the primary components found in the oil-resin. By subjecting the GC/MS data to chemometric analysis, a PCA plot was produced which revealed distinct sample groups that largely corresponded to the origin of the samples. The PCA was also used to detect outliers, which were subjected to additional investigation. A SFC/MS method was developed to detect and identify the addition of soybean oil. Samples which were determined to be outliers by the PCA were subjected to SFC/MS analysis. All of the outliers were found to be adulterated with soybean oil. By combining multiple analytical techniques, the quality of the oil-resin and its adulteration can easily be determined. Overall, by utilizing these methods, the quality and safety of copaiba oil-resin can be improved. Perhaps, in the future, the combination of the described methods can also be used to assess the quality of other essential oils.

## Figures and Tables

**Figure 1 plants-12-01619-f001:**
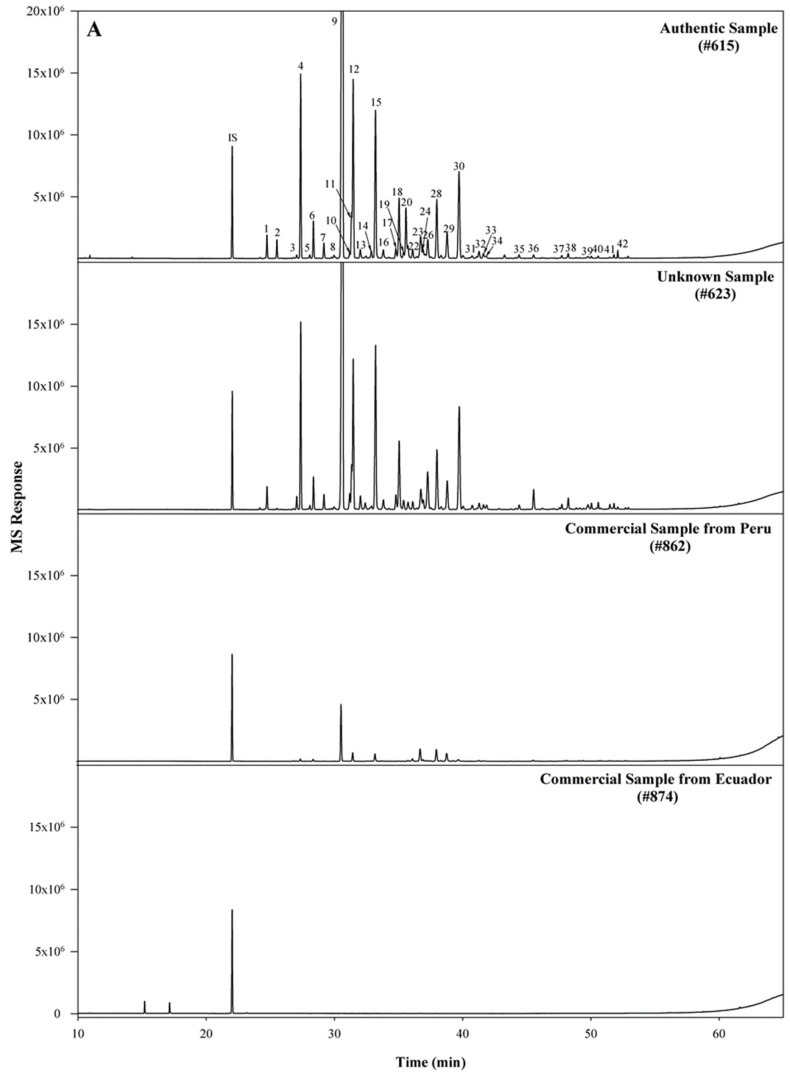
(**A**) Representative total ion chromatograms for: authentic (#615), unknown sample (#623), and commercial products (#862 from Peru and #874 from Ecuador). Compound identification is indicated in Table 1. (**B**) Representative total ion chromatograms for unknown samples collected from different locations in Brazil (#888, Tapuua; #891, Apui; #894, Parintins; #897, Labrea).

**Figure 2 plants-12-01619-f002:**
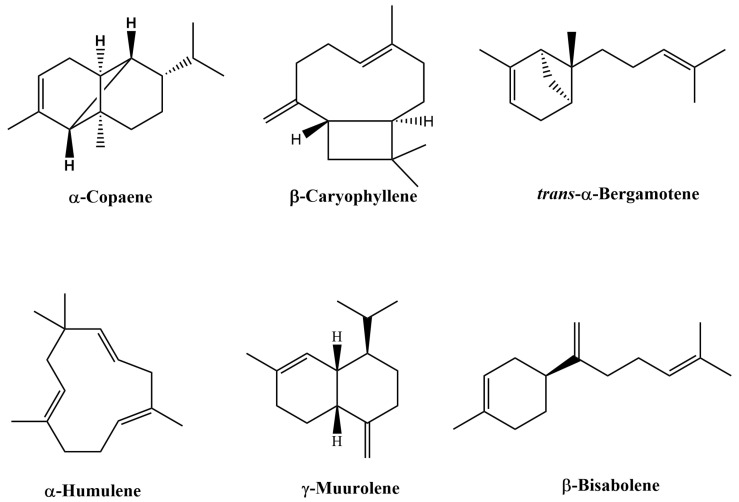
Characteristic compounds present in copaiba oil-resin.

**Figure 3 plants-12-01619-f003:**
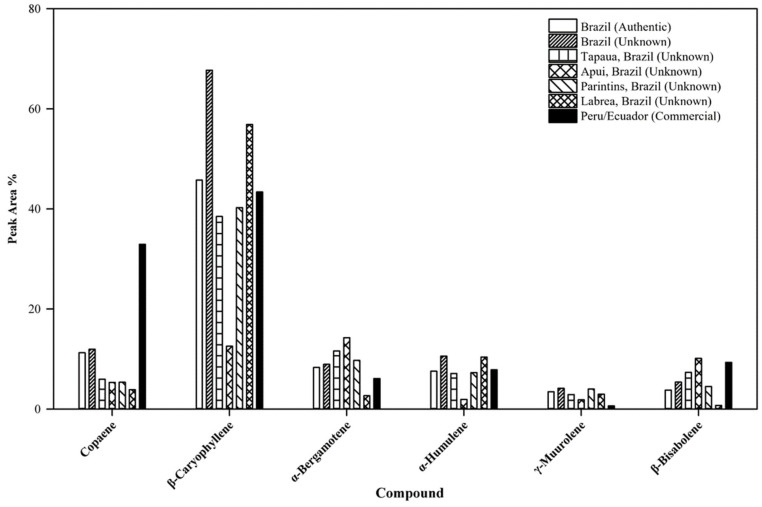
Comparison of major compounds in each sample group.

**Figure 4 plants-12-01619-f004:**
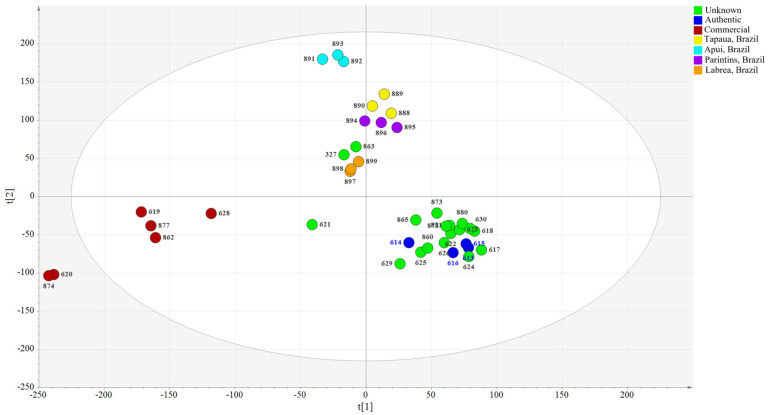
PCA score plot of all the samples.

**Figure 5 plants-12-01619-f005:**
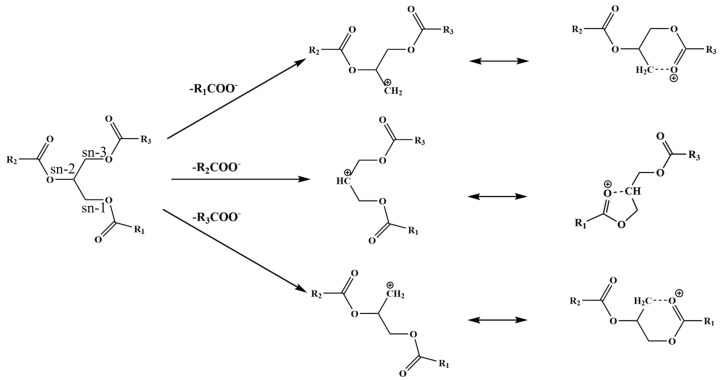
Schematic representation of the fatty acids from the sn-1, sn-2, and sn-3 positions on the glycerol backbone by SFC/MS with ESI positive ionization.

**Figure 6 plants-12-01619-f006:**
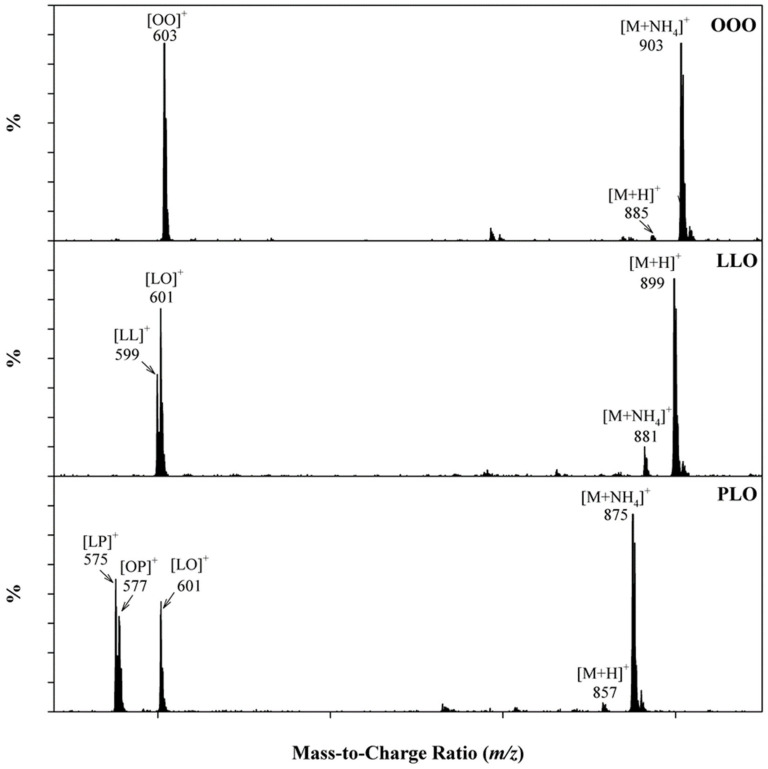
Representative ESI(+) MS spectra to demonstrate the identification of triglycerides with different acyls on the glycerol backbone.

**Figure 7 plants-12-01619-f007:**
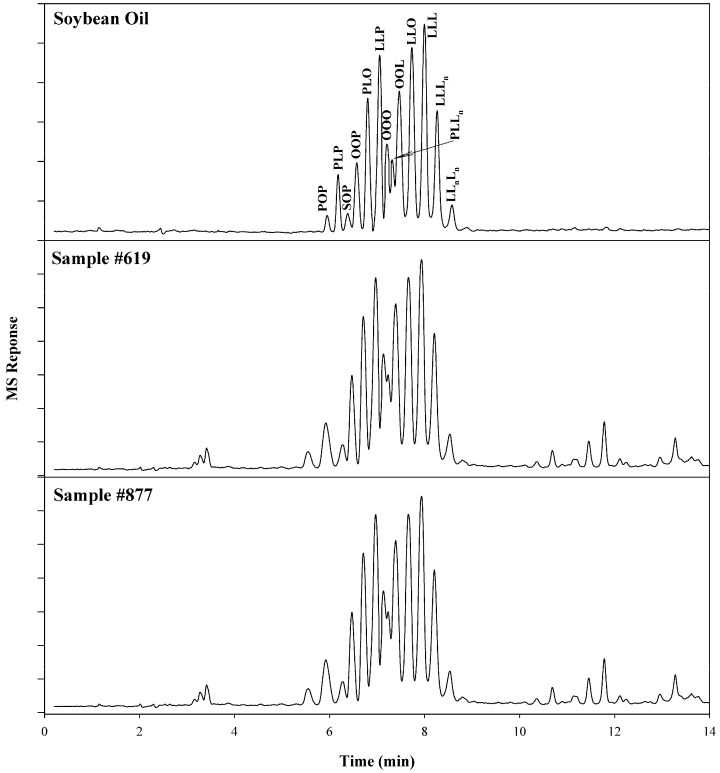
SFC/MS total ion chromatograms of a soybean oil standard and two adulterated commercial samples containing soybean oil.

**Table 1 plants-12-01619-t001:** Tentative compound identification and range (average) of constituent concentration (peak area %) of copaiba oil-resin.

No.	Compound *	RRI^Cal^	RRI^Lit^	Authentic Sample	Unknown Sample	Unknown Sample with Known Location	Commercial Sample
1	δ-EIemene	1342	1338	0.84–1.92 (1.15)	0.00–2.25 (1.00)	0.05–0.96 (0.35)	0.00–0.07 (0.04)
2	α-Cubebene	1355	1351	0.65–1.45 (0.83)	0.00–1.91 (0.73)	0.20–0.59 (0.35)	0.00–4.65 (0.78)
3	α-Ylangene	1379	1375	0.13–0.25 (0.18)	0.09–0.63 (0.27)	0.04–0.16 (0.08)	0.00–0.18 (0.08)
4	α-Copaene	1384	1376	7.33–11.25 (9.28)	3.66–11.3 (8.66)	3.55–5.96 (5.01)	0.00–32.93 (5.95)
5	7-epi-Sesquithujene	1395	1391	0.12–0.21 (0.15)	0.03–1.95 (0.28)	0.07–0.26 (0.20)	0.00–0.16 (0.06)
6	β-Elemene	1399	1390	1.43–1.94 (1.81)	0.00–2.23 (1.35)	1.10–1.64 (1.30)	0.00–6.20 (1.42)
7	Cyperene	1409	1398	0.64–0.80 (0.75)	0.16–0.73 (0.59)	0.25–0.49 (0.39)	0.00–0.02 (0.00)
8	α-Cedrene	1419	1411	0.15–0.20 (0.19)	0.00–0.21 (0.16)	0.00–0.32 (0.15)	0.00–0.03 (0.01)
9	β-Caryophyllene	1427	1419	38.58–45.77 (43.71)	35.49–67.71 (43.11)	11.61–56.87 (36.34)	0.00–43.40 (20.87)
10	cis-β-Copaene	1433	1432	0.22–0.31 (0.24)	0.12–0.94 (0.34)	0.04–0.15 (0.09)	0.00–0.05 (0.03)
11	γ-Elemene	1435	1436	1.46–1.86 (1.76)	0.00–2.18 (1.42)	0.00–0.96 (0.21)	0.00
12	trans-α-Bergamotene	1437	1434	7.55–8.31 (7.78)	3.46–8.94 (6.87)	2.29–14.24 (9.23)	0.00–6.09 (2.81)
13	Aromandendrene	1443	1441	0.37–0.44 (0.40)	0.12–0.67 (0.43)	0.13–0.54 (0.32)	0.00–0.08 (0.03)
14	trans-β-Farnesene	1454	1456	0.30–0.43 (0.34)	0.00–0.46 (0.22)	0.04–0.60 (0.36)	0.00–0.20 (0.10)
15	α-Humulene	1458	1454	6.26–7.56 (7.10)	6.07–10.57 (7.67)	1.91–10.37 (6.50)	0.00–7.84 (3.57)
16	allo-Aromadendrene	1465	1460	0.46–0.50 (0.48)	0.20–0.71 (0.48)	0.00–0.46 (0.35)	0.00–2.95 (0.49)
17	Cadina-1(6),4-diene	1477	1463	0.50–0.65 (0.57)	0.18–1.10 (0.53)	0.18–0.47 (0.29)	0.00–0.19 (0.05)
18	γ-Muurolene	1480	1479	2.97–3.46 (3.08)	1.06–4.06 (3.25)	1.80–4.01 (2.85)	0.00–0.63 (0.24)
19	α-Amorphene	1484	1484	0.25–0.34 (0.29)	0.00–0.47 (0.09)	0.00–0.32 (0.08)	0.00–0.10 (0.02)
20	Germacrene D	1486	1481	1.34–2.73 (1.84)	0.00–2.46 (0.99)	0.00–0.72 (0.14)	0.00–1.96 (0.33)
21	9-epi-β-Caryophyllene	1488	1466	0.54–0.69 (0.59)	0.00–0.76 (0.50)	0.00–2.74 (0.77)	0.00–0.69 (0.27)
22	β-Eudesmene	1492	1490	0.47–0.65 (0.57)	0.26–0.97 (0.47)	0.41–2.36 (1.29)	0.00–11.04 (2.42)
23	Viridiflorene	1499	1496	1.49–1.64 (1.54)	0.00–2.01 (1.02)	0.00–0.56 (0.09)	0.00–0.19 (0.03)
24	α-Selinene	1501	1498	0.52–0.61 (0.57)	0.31–1.02 (0.54)	0.41–2.15 (1.19)	0.00–6.19 (1.39)
25	Bicyclogermacrene	1504	1500	0.10–0.22 (0.14)	0.00–0.22 (0.05)	0.00–0.06 (0.01)	0.00–0.43 (0.16)
26	α-Muurolene	1506	1500	0.98–1.40 (1.11)	0.00–2.54 (1.29)	0.00–1.90 (0.87)	0.00–0.44 (0.16)
27	β-Bisabolene	1513	1505	2.78–3.78 (3.13)	1.59–5.38 (3.16)	0.63–10.11(5.48)	0.00–9.31 (4.12)
28	β-Curcumene	1516	1515	0.10–0.18 (0.15)	0.08–0.27 (0.16)	0.00–0.91 (0.33)	0.00–0.51 (0.20)
29	γ-Cadinene	1521	1513	0.95–1.57 (1.27)	0.66–2.09 (1.54)	1.44–4.88 (2.49)	0.00–7.24 (3.64)
30	δ-Cadinene	1534	1523	4.41–5.65 (4.99)	0.17–6.91 (4.76)	3.99–7.28 (5.74)	0.00–8.10 (2.02)
31	trans-Cadina-1,4-diene	1541	1534	0.11–0.19 (0.14)	0.07–0.36 (0.18)	0.09–0.19 (0.15)	0.00–0.31 (0.06)
32	α-Cadinene	1546	1538	0.32–0.42 (0.35)	0.16–0.59 (0.37)	0.00–0.69 (0.24)	0.00–0.68 (0.12)
33	trans-γ-Bisabolene	1550	1531	0.20–0.33 (0.24)	0.10–0.49 (0.25)	0.00–1.73 (0.72)	0.00–0.43 (0.15)
34	Selina-3,7(11)-diene	1551	1564	0.00–0.17 (0.11)	0.00–0.35 (0.13)	0.29–1.11 (0.53)	0.00–0.10 (0.02)
35	Caryophyllenyl alcohol	1578	1572	0.14–0.19 (0.15)	0.08–0.58 (0.21)	0.12–0.81 (0.39)	0.00–0.11 (0.05)
36	Caryophyllene oxide	1590	1583	0.18–0.23 (0.21)	0.00–3.53 (0.67)	0.00–0.37 (0.23)	0.00–1.45 (0.64)
37	Gleenol	1619	1587	0.13–0.22 (0.17)	0.00–0.35 (0.19)	0.00–0.34 (0.22)	0.00–0.11 (0.02)
38	Junenol	1627	1619	0.10–0.31 (0.19)	0.10–0.99 (0.35)	0.37–0.74 (0.50)	0.00–1.11 (0.22)
39	τ-MuuroloI	1652	1642	0.06–0.16 (0.09)	0.00–0.73 (0.19)	0.11–0.79 (0.43)	0.00–0.40 (0.08)
40	δ-Cadinol	1657	1646	0.04–0.12 (0.07)	0.00–0.66 (0.17)	0.12–0.73 (0.50)	0.00–0.07 (0.02)
41	α-Cadinol	1686	1654	0.13–0.23 (0.17)	0.07–0.25 (0.18)	0.11–0.21 (0.15)	0.00–0.03 (0.01)
42	Eudesm-7(11)-en-4-ol	1691	1700	0.14–0.25 (0.21)	0.07–0.60 (0.22)	0.00–0.70 (0.43)	0.00–0.06 (0.03)
43	16-Kaurene	2035	2043	0.00	0.00–0.22 (0.01)	0.00–0.40 (0.16)	0.00–0.15 (0.03)
44	Manool	2038	2057	0.00	0.00–0.33 (0.03)	0.00–0.11 (0.04)	0.00
45	Kolavelool	2043	-	0.00	0.00–1.34 (0.14)	0.00–0.11 (0.03)	0.00
46	Kolavenol	2323	2297	0.00	0.00–1.07 (0.09)	0.00–0.34 (0.10)	0.00–0.18 (0.03)
47	Methyl kolavenate	2395	-	0.00	0.00–0.52 (0.03)	0.00–0.10 (0.03)	0.00–3.68 (1.30)

*: Names of compounds were provided according to the NIST mass spectral library. The isomer was specified when possible. RRI^Cal^: relative retention indices calculated against n-alkane. RRI^Lit^: relative retention indices data from the literature. -: compound not detected or only contain trace amount.

**Table 2 plants-12-01619-t002:** Sample Information.

NCNPR Code	Information on Label	Country of Origin
**Authentic Samples**		
613	Raw *Copaiba*	Brazil
614	Raw *Copaiba*	Brazil
615	Raw *Copaiba*	Brazil
616	Raw *Copaiba*	Brazil
860	Raw *Copaiba*	Brazil
**Unknown Samples**		
327	*Copaifera officinalis*	N/A
617	Raw VC *Copaiba*	Brazil
618	*Copaiba*	Brazil
621	Oil of *Balsam Copaiba*	N/A
622	*Copaiba*	N/A
623	*Copaifera officinalis*	N/A
624	*Copaiba*	Brazil
625	*Copaiba*	N/A
626	*Copaiba*	N/A
629	*Copaifera*	Brazil
630	*Copaiba*	N/A
863	*Copaiba balsam*	N/A
865	*Copaiba*	N/A
871	*Copaiba*	N/A
873	*Copaiba*	N/A
878	Raw *Copaiba*	N/A
880	*Copaifera officinalis*	N/A
888	*Copaifera officinalis*	Tapaua, Brazil
889	*Copaifera officinalis*	Tapaua, Brazil
890	*Copaifera officinalis*	Tapaua, Brazil
891	*Copaifera officinalis*	Apui, Brazil
892	*Copaifera officinalis*	Apui, Brazil
893	*Copaifera officinalis*	Apui, Brazil
894	*Copaifera officinalis*	Parintins, Brazil
895	*Copaifera officinalis*	Parintins, Brazil
896	*Copaifera officinalis*	Parintins, Brazil
897	*Copaifera officinalis*	Labrea, Brazil
898	*Copaifera officinalis*	Labrea, Brazil
899	*Copaifera officinalis*	Labrea, Brazil
**Samples from Commercial Sources**		
619	*Copaiba*	Peru
620	*Copaifera*	Ecuador
628	*Copaiba*	Peru
862	*Copaiba*	Peru
874	*Copaifera*	Ecuador
877	*Copaiba*	Peru

Bold texts indicate different sample groups.

## Data Availability

The data that support the findings of this study are available from the corresponding authors upon request.

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
