# Peer review of "Chemical Characterization and Quality Assessment of Copaiba Oil-Resin Using GC/MS and SFC/MS"

_plants, 2023, doi:10.3390/plants12081619_

Round 1

Reviewer 1 Report

Dear authors,

let me congratualte you on a well done job. The articule was interesting to read, for me as researcher. The idea is clear and well accoplished. the object is well known in the world of essential oils but researchers still have a lack of information about the authentic composition of Copaiba oilraisin. The adulteration problem is also very important and it needs to be widly proven and discused. 

Methods for your research are picked very strategic and lead to the results that are hard to deny. 

The only thing that I would suggest to change in Your manuscript is this abnormally huge table of chemical compounds of all the samples. Because you devided all samples in several groups, it would be enought to give a range of composition for each compound in each group.  Another obcion is to move all this huge table to supplementary material and leave in the text the smaller version with only those six characteristic compounds. Also you declare that all smaples were analysed in triplicate, so compositions should be presented as average means with standard deviations. 

Overal it is a very interesting paper and will find his audience in scientific world.

Reviewer 2 Report

The manuscript entitled „Chemical Characterization and Quality Assessment of Copaiba Oil-Resin Using GC/MS and SFC/MS” presents a MS and PCA-based method of evaluation of quality of natural Copaiba oil. The results are clearly presented with only minor issues:

-abstract – it should be rewritten (as well as the conclusions). Too many repetitions from the main text.

- Introduction should include brief discussion of other PCA-based method for QA. Especially it is important to show the extent this work is original besides the material investigated

- p3. section 2.1. It seems that this section repeats mainly the data showed in Table 1. Maybe this section should be shortened and only general trend described.

Table 1 – it is not clear which data are “authentic” and “unknown”, two horizontal lines spanning the correct range would be helpful

- fonts in Fig 2 and 3 should be bigger (legends and axis)

- sn-1, sn-2 and so an are not explained, alternatively standard indications of structure should be used

p9.l162- the introduction of PCA is not necessary, it is well established statistical method.

p. 11 l.235 – R1R1  -> R1R2

Table 2 NCNPR – abbreviation not explained

p14 l.278 – numbers of atoms not subscripted

p14  UPC--> UPLC

p15. Conclusion  section should be rewritten, there are too many reused sentences from previous sections (especially it is too similar with the abstract). Some discussion about other similar methods should be included.

Reviewer 3 Report

The paper contains a characterization of forty copaiba oil-resin samples analyzed by GC/MS, supercritical chromatography/mass spectrometry and PCA for grouping.  The paper presents particular scientific interest from point of view of identification of components, standardization, and adulteration of  Copaiba oil-resin. The authors discriminate six characteristic substances of Copaiba oil-resin. 

Some corrections should be done to improve this manuscript.

I propose to give structures of these six components to improve the visualization of the paper.

Secondly, It is known that GC/MS gives abundances of components, not concentrations. Please correct it in the manuscript.

Chapter 2.1. It is difficult a little bit to perceive information. I wish the authors would do more clear accent on authentic samples and unknown samples.... I mean it is necessary to add an introductory sentence that gives the short characteristic of samples analyzed in this paper. This information is only in chapter 3

line 43-44 Traditionally, copaiba oil has been used for the treatment of skin disorders and inflammation.

line 100-101 With this in mind, our goal was to evaluate the chemical composition and variation of samples obtained from different geographic regions

Reviewer 4 Report

dear Editor,

I believe that the document in its current state meets the quality requirements of the journal l, however, I have a few comments stated below:

-Line 265 „were authenticated (passed third party quality testing at the time 265 of collection)“ could you please add some additional data about this authentification or provide it as a reference?

-Please include all used chemicals in the Chemicals and reagents list and provide purity and producer

-the conclusion section is too long (with many repeated dana from the main section)

-3.4. Gas Chromatography/Mass Spectrometry (GC/MS) analyses. was the quantification of the compounds carried out and how?
